



# Exploring the footprint representation of microwave radiance observations in an Arctic limited-area data assimilation system

Máté Mile[1,2], Stephanie Guedj[1], and Roger Randriamampianina[1]

[1]The Norwegian Meteorological Institute, Development Centre for Weather Forecasting, Oslo, Norway
[2]University of Oslo, The Faculty of Mathematics and Natural Sciences, Department of Geosciences, Oslo, Norway

**Correspondence:** Máté Mile (matem@met.no)

**Abstract.** The microwave radiances are key observations especially over data sparse regions for operational data assimilation in numerical weather prediction (NWP). An often applied simplification is that these observations are used as point measurements, however, the satellite field-of-view may cover many grid points of high-resolution models. Therefore, we examine a solution in high-resolution data assimilation to better account for the spatial representation of the radiance observations. This

solution is based on a footprint operator implemented and tested in the variational assimilation scheme of the AROME-Arctic (Application of Research to Operations at Mesoscale - Arctic) limited-area model. In this paper, the design and technical challenges of the microwave radiance footprint operator are presented. In particular, implementation strategies, the representation of satellite field-of-view ellipses, and the emissivity retrieval inside the footprint area are discussed. Furthermore, the simulated brightness temperatures and the sub-footprint variability are analysed in a case study indicating particular areas where the use

of the footprint operator is expected to provide significant added value . For radiances measured by the Advanced Microwave Sounding Unit-A (AMSU-A) and Microwave Humidity Sounder (MHS) sensors, the standard deviation of the observation minus background (OmB) departures are computed on a short period in order to compare the statistics of the default and the implemented footprint observation operator. For all operationally used AMSU-A and MHS channels, it is shown that the standard deviation of OmB departures is reduced when the footprint operator is applied.

## 1 Introduction

The satellite radiances are essential components of operational state-of-the-art data assimilation (DA) systems. Microwave sounders and their tropospheric sounding channels are dominant contributors providing vital atmospheric observations over data sparse regions like the Arctic (Bormann et al., 2017; Randriamampianina et al., 2019). However, in limited-area models

(LAMs), the radiance DA is challenging for several reasons. For example, the irregular data sample at the different analysis times can be mentioned which leads to specific treatment of the variational bias correction (Randriamampianina et al., 2011; Lindskog et al., 2012; Cameron and Bell, 2016; Benáček and Mile, 2019). In most operational NWP systems, radiance DA





was developed and implemented in coarse-resolution models. It makes the high-resolution DA systems suboptimal when the satellite products are used as point measurements and the representation error is not taken into account. Despite all these

challenges, satellite radiances and in particular microwave data have positive impact on analyses and forecasts in LAMs (e.g. Randriamampianina et al. (2019); Lindskog et al. (2021)). In general, the use of radiance measurements is more straightforward over open ocean in clear-sky conditions and the complexity of their assimilation grows with the increased sensitivity to the non-ocean surface and over areas affected by precipitation and clouds.

Over sea, the available water emissivity models at microwave frequencies are accurate (e.g., Liu et al. (2011)) and have been

successfully employed by many operational DA centres due to the fact that the sensitivity to the surface is relatively small. Over land and sea ice, the surface emissivity and skin temperature are more uncertain and dependent on various surface parameters. The surface emissivity atlases (e.g., Aires et al. (2011)) and dynamic emissivity retrievals (Karbou et al., 2006, 2014; Baordo and Geer, 2016) facilitate the use of surface-sensitive microwave radiances over non-ocean surfaces. The dynamic emissivity method is based on a surface emissivity retrieval from window channel observations and also using background information of the NWP model for both temperature and humidity-sounding channels. This method enabled further extension of the assim-

ilation of surface-sensitive microwave data over land as well as the extension of the use of humidity-sounding channels over sea-ice and snow-covered surfaces (Karbou et al., 2010; Di Tomaso et al., 2013). Another major advancement was achieved by the implementation of the all-sky radiance DA (Bauer et al., 2010; Geer et al., 2014) which significantly increased the amount of active radiance observations sensitive to cloud and precipitation. However, these developments still have limitations where stronger sensitivity to land and sea-ice surfaces are present thus radiance data need to be rejected. At the European Centre for

Medium-Range Weather Forecasts (ECMWF), further developments have been initiated to appropriately assimilate radiance data over complex or mixed surfaces which is called the all-sky all-surface approach (Geer et al., 2022). Recently, a better representation of satellite radiance measurements in DA was studied by Bormann (2016, 2017), and Shahabadi et al. (2020) through the implementation of slant-path observation operators.

The above-mentioned radiance DA developments initiated in global models have been gradually implemented into many regional DA systems (e.g., Schwartz et al. (2012); Storto and Randriamampianina (2010); Lindskog et al. (2021)), however, the all-sky assimilation and the use of low-peaking channels remain still subject of research at many centres. It is worth mentioning that radiance observations assimilated in global models have significant impact on the LAM systems through the lateral boundary conditions (Randriamampianina et al., 2021). In high-resolution DA, the satellite field-of-view (FOV) is

inadequately represented by the existing observation operator which leads to increased representation error. The representation error (Janjić et al., 2017) by definition is an error component which might come from the unresolved scales and processes of the high-resolution observations. However, in high-resolution models, the opposite scale mismatch often occurs when unobserved scales and processes of the NWP model cause the representation error in DA (Marseille and Stoffelen, 2017; Mile et al., 2021). In high-resolution LAMs, the footprint operator aims to reduce the error related to spatial representation error by avoiding the

correction of the smallest scales of the model in the DA procedure. The footprint featured operators have been implemented in different geoscientific applications, for example, in regional ice analysis system of Environment Climate Change Canada (Buehner et al., 2013; Carrieres et al., 2017) or in the recently studied scatterometer and Aeolus data assimilation in regional





NWP (Mile et al., 2021, 2022). In radiance data assimilation, the importance of the satellite footprint representation has been recognised as well. Kleespies (2009) showed that the aggregation of model surface quantities over the satellite footprint

of the microwave radiances might be important when the surface parameters vary substantially. Probably for the first time, Duffourg et al. (2010) studied a radiance footprint operator of infrared radiances for a convective-scale DA system and over the Mediterranean domain. Recently, Chen et al. (2022) proposed another approach to reduce representation error in microwave radiance data assimilation by applying the so-called remapping technique based on a convolutional neural network.

In this study, we aim to explore the importance of the footprint representation of microwave radiance observations focusing

on an Arctic NWP domain and using the AROME-Arctic mesoscale model (Randriamampianina et al., 2019). Section 2 explains the data and methods of this study i.e., the microwave radiances and the NWP model configurations. Section 3 describes the challenges of the footprint representation and the implemented footprint operator. In Section 4, a case study is analysed in order to assess the importance of the footprint representation by examining the variability in simulated brightness temperatures inside the satellite FOV. Section 5 presents data assimilation diagnostics and Section 6 draws the conclusion and

future developments.

## 2 Data

### 2.1 The microwave sounding measurements

The AMSU-A is designed to measure temperature profiles by a cross-track scanning radiometer. The AMSU-A sensor detects radiances in 15 discrete frequencies from 23.8 to 89 GHz having 11 sounding channels near oxygen absorption band for tem-

perature sounding and 4 other channels more sensitive to the surface. At each channel frequency, the instrument provides 30 observations from -48° to +48° degrees with respect to the nadir direction. The geometric resolution of a nadir circular instantaneous FOV (IFOV) has roughly 48 km diameter size and it's gradually increased towards the edges of the swath to roughly 147 km across- and 79 km along-track resolutions forming an elliptical IFOV. The MHS is designed for tropospheric humidity retrievals by a self-calibrating, cross-track scanning, five-channel microwave radiometer. The MHS scans the atmosphere in

frequencies from 89 to 190 GHz with 3 channels dedicated to the strong 183.3 water vapour line and with 2 others called window channels at 89 and 157 GHz. The instrument produces 90 observations of each channel with the approximate nadir resolution of 16 km at a nominal altitude of 833 km. The footprint size increases towards the edges of the swath like in the case of all cross-track scanning satellites and reaches 53 km across- and roughly 27 km along-track resolution. These microwave instruments are carried on board EUMETSAT's (European Organisation for the Exploitation of Meteorological Satellites) Metop

(Meteorological Operational satellite) and NOAA's (National Oceanic and Atmospheric Administration) satellites. Examples of AMSU-A and MHS FOV footprint ellipses are shown near the edge of the swath (Figure 1a) and around the nadir (Figure 1b) position over Northern Scandinavia.

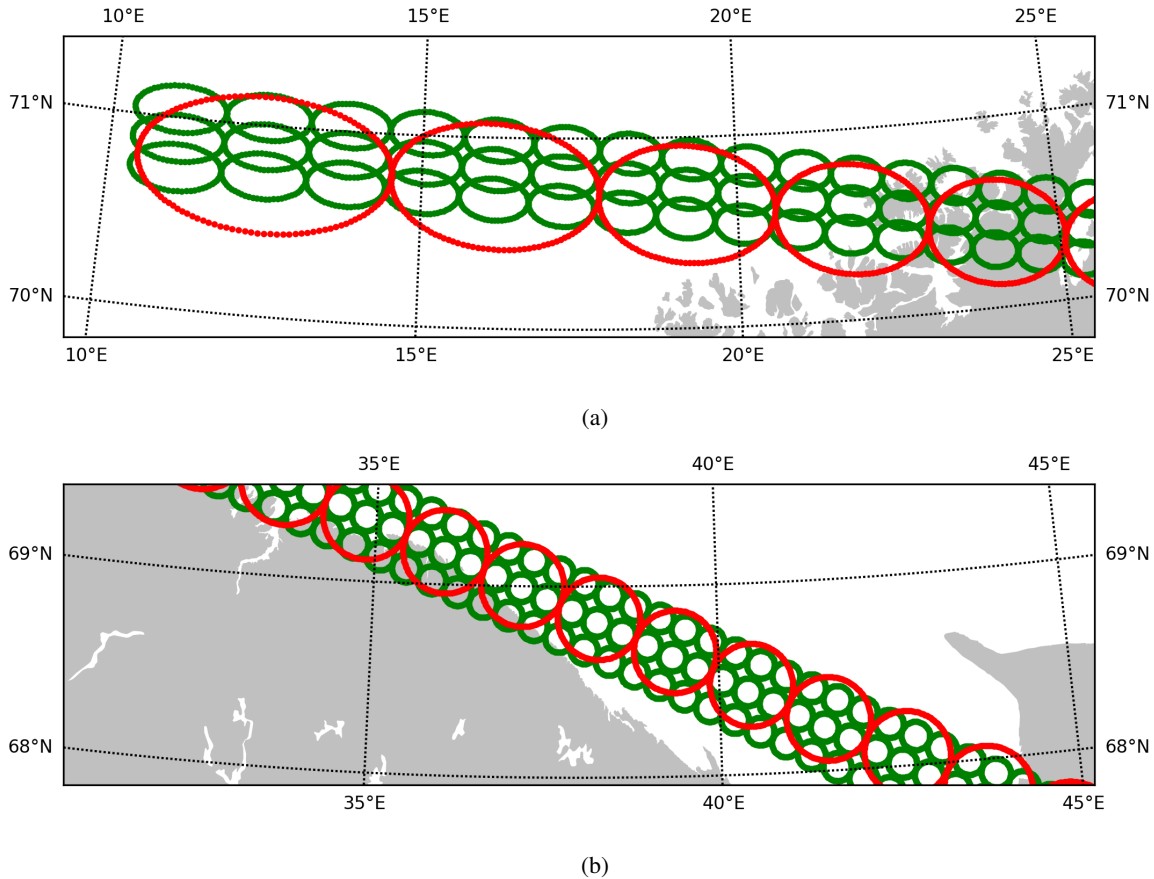

**Figure 1.** The FOV footprint ellipses for AMSU-A (red) and MHS (green) sensors on METOP-C satellite near the edge of the swath (**a**) and around the nadir (**b**). Sample taken from measurement on 20 March, 2020, 08:28 UTC.

## 2.2 AROME-Arctic operational assimilation system

The AROME-Arctic (AA) model (Müller et al., 2017; Randriamampianina et al., 2019) is one of the regional NWP model used at the Norwegian Meteorological Institute (MET Norway). It has been operational since November 2015 and covers parts of the Norwegian, Barents and Greenland seas, the Svalbard archipelago, and northern Scandinavia. In our study, the model code version 43h2.1 is used in both the reference and the radiance footprint operator implementation runs. The AA uses 2.5 km horizontal resolution and 65 vertical levels which is the current operational setup. The AA system assimilates all available conventional and many non-conventional observations from polar-orbiting meteorological satellites. Regarding the use of microwave radiance observations, AA uses data from Metop-B, Metop-C, NOAA-18, and NOAA-19 satellites. In the observation operator of AA assimilation system, the radiative transfer model (RTTOV) version 11.2 is incorporated (NWP SAF, 2013). The radiance observations are not assimilated in full resolution due to their correlated observation errors and therefore satellite data are undergo a two-step thinning procedure in which the minimum distance is ensured first, then the thinning to an





average distance is done as a second step. The minimum and average thinning distances are 60 and 80 km for AMSU-A and

40 and 80 km for MHS radiances respectively. The microwave radiances are assimilated operationally in clear-sky conditions using the following channels 5, 6, 7, 8, and 9 from the AMSU-A and 3, 4, and 5 from the MHS instruments. Regarding the blacklisting of these microwave sensors, the operational system rejects the first and last 3 and 9 IFOVs of AMSU-A and MHS respectively. The AA assimilation system utilises the three-dimensional variational (3D-Var) scheme (Fischer et al., 2005) with increased assimilation cycle frequency producing 8 analyses per day. The background error covariance matrix was computed

considering a dataset for four seasons from the downscaled ECMWF's ensemble data assimilation (EDA) runs. In this study, we do not aim to modify the predefined background and observation errors, although, the use of the newly developed footprint operator might prefer the recomputation or the tuning of these errors and adjustment of the quality control procedure.

## 3   Methods

### 3.1   The choice of the radiance footprint operator

By the default observation operator, radiance data just like any other observation types are assimilated as point observations and a horizontal interpolator (mostly bilinear interpolation using 4 neighbouring grid-points) is employed to generate the model equivalent at the observation location. In order to take into account the satellite footprint, this procedure needs to be replaced or extended. Generally, a footprint operator computes the average of high-resolution model quantities over the footprint of the satellite sensor.

For the implementation of the radiance footprint operator, different strategies and solutions can be considered (see for instance Duffourg et al. (2010)). One solution would be to completely replace interpolation with averaging technique using the model quantities around the observation location i.e., does the averaging in model space. Such implementation would be cost efficient due to the fact that only one averaged model information is needed as input in the RTTOV. However, it still requires an extended computational domain or computational halo to access the increased number of grid points (see the example in

Mile et al. (2021)). Moreover, it has major drawbacks for radiance data like the inappropriate footprint representation and the neglected effects of the incorporated RTTOV in the observation operator. Another more expensive and probably more sophisticated implementation might be to apply the averaging in observation space after the interpolation. In this case, several additional operator points around the actual geolocation of the data are needed to construct the instrument footprint. The AA model source code as part of the ARPEGE/IFS (Action de Recherche Petite Echelle Grande Echelle/Integrated Forecasting

System) model family has an already existing feature for such developments using the so-called 2D-GOM-arrays. These arrays can store additional model profiles (model-equivalent profiles) for complex observation operators like the slant-path radiance operator (Bormann, 2017) in ECMWF's IFS global model, the radar reflectivity operator in AROME-France (Wattrelot et al., 2013), the slant total delay observation operator (de Haan et al., 2020), or the Aeolus footprint operator (Mile et al., 2022) in HARMONIE-AROME (HIRLAM ALADIN Research on Mesoscale Operational NWP In Europe – convective-scale

forecasting system - AROME). Considering the footprint representation in observation space, there can still be different options where to perform the average in the footprint operator related to the trade-off between complexity and computational cost. One





option would be to average the model quantities (over the IFOV) before the actual call of the RTTOV neglecting the non-linear behaviour of the radiative transfer and saving computer resources by executing RTTOV only once. Another, more advanced, solution would be to run RTTOV for all model profiles (inside the IFOV) contributing to the footprint operator independently
and average them afterwards taking into account the non-linear effects of the RTTOV.

In this study, our aim is to explore the more advanced solution for the microwave radiance footprint operator, therefore, the averaging in observation space after the RTTOV calls has been implemented. The implementation involves many technical challenges which are discussed in the following sections.

### 3.2 The representation of IFOV and the sampling of the footprint operator points

Both AMSU-A and MHS are across-track scanning instruments with similar scanning geometry properties. The antenna beamwidth of each AMSU-A channel is around 57.6 milliradians (3.3°) making a circular IFOV and the sampling angular interval is roughly 58.18 milliradians (3.3333°). The distance between satellite scans is around 52.69 km. Regarding MHS sensor, the beamwidth of each channel is approximately 19.2 milliradians (1.1°) and the sampling angular interval is around 19.39 milliradians (1.1111°). The distance between MHS consecutive scans is roughly 17.56 km. The IFOV area for both in-
struments at nadir forms a perfectly circular footprint with the diameter of 48 and 16 km for AMSU-A and MHS respectively. When the sensor looks towards the edge of the scan, the footprint area elongates notably in the cross-track while slightly in the along-track directions as well. The footprint ellipses of such radiance instruments can be described and defined by their minor and major axes relative to the nadir viewing angle of the satellite. The viewing angle of the pixel centre and the vertices of a given IFOV ellipse can be expressed as


$$
\quad \alpha_s = \frac{\delta_{step}}{2} + (p-1) * \delta_{step} - \epsilon \quad (1) \quad \alpha_e = \frac{\delta_{step}}{2} + (p-1) * \delta_{step} + \epsilon \quad (2) \quad \alpha = \frac{\delta_{step}}{2} + (p-1) * \delta_{step}, \quad (3)
$$

where the $\delta_{step}$ and $\epsilon$ stand for the sampling angular interval (i.e., centre to centre IFOV step angle) and the half of the IFOV width respectively. $p$ means the pixel number, furthermore, $\alpha$, $\alpha_s$, $\alpha_e$ stand for the viewing or scanning angle of the pixel centre and the two cross-track (i.e., start and end) vertices of the IFOV ellipse respectively. These $\alpha$ viewing angles can be converted
in order to obtain the scanning angle (hereafter $\gamma$) on the Earth's frame as the following


$$
\gamma_s = \arcsin\left(\left(\frac{R_E + H_{sat}}{R_E}\right) * \sin(\alpha_s)\right) - \alpha_s, \; \gamma_e = \arcsin\left(\left(\frac{R_E + H_{sat}}{R_E}\right) * \sin(\alpha_e)\right) - \alpha_e, \; \gamma = \arcsin\left(\left(\frac{R_E + H_{sat}}{R_E}\right) * \sin(\alpha)\right) - \alpha,
$$

$$
(4) \qquad\qquad (5) \qquad\qquad (6)
$$





where the $R_E$ is the Earth's radius and $H_{sat}$ is the altitude of the satellite at a given geographical location. On Figure 2, an illustration of cross-track scanning field-of-view ellipses and the introduced scanning angles are presented.

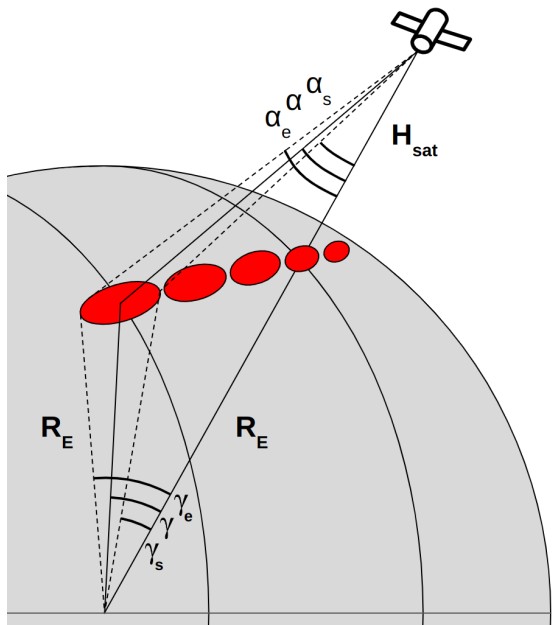

**Figure 2.** Schematic satellite field-of-view ellipses and the scanning angles of the IFOV centre and ellipse vertices for the first IFOV of the swath.

The length of the major axis of an IFOV ellipse on the Earth's surface, therefore, can be expressed as follows

$$d_{major} = (\gamma_e - \gamma_s) * R_E, \tag{7}$$

and the satellite's slant-path can also be expressed using the law of sines to compute the minor axis of the IFOV ellipse as follows


$$d_{path} = \frac{R_E * \sin(\gamma)}{\sin(\alpha)} \tag{8} \qquad d_{minor} = d_{path} * 2\epsilon. \tag{9}$$

Using $d_{major}$ and $d_{minor}$, the actual size of the IFOV can be calculated and used for the footprint operator. The spatial sampling of the footprint operator points (hereafter FOPs) should reflect the horizontal resolution of the applied NWP model 180  taking into account the characteristics of the ground-projected IFOV. In our implementation, the FOPs and the geolocations are taken equidistantly along the major and minor axes of an IFOV ellipse, however, it does not result in equal sampling distances





within the full satellite scan due to the changing size of the IFOV. Additionally, the same number of FOPs is used in all IFOVs which is a technical constraint in the current use of 2D-GOM-arrays making the sampling more appropriate for IFOVs with small satellite viewing angles. Nevertheless in this implementation, the sampling distance on the average is close to the

model resolution namely 2.51 and 2.42 km for AMSU-A and MHS respectively. The FOPs are selected in variable directions around the observation location in order to avoid high-resolution sampling at the centre of the IFOV (see the operator sampling distribution on Figure 3 and 4). A rotation of the FOPs is also needed to align the IFOV ellipses to the orientation of satellite scanlines or scanning directions. To this, the distance of all FOPs from the observation location is computed and rotated by the satellite azimuth as follows

$$D_{rot} = \begin{bmatrix} \cos(\phi - 90°) & -\sin(\phi - 90°) \\ \sin(\phi - 90°) & \cos(\phi - 90°) \end{bmatrix} \times \begin{bmatrix} \varphi_{foop}^1 - \varphi_{obs} & \varphi_{foop}^2 - \varphi_{obs} & \cdots & \varphi_{foop}^m - \varphi_{obs} \\ \lambda_{foop}^1 - \lambda_{obs} & \lambda_{foop}^2 - \lambda_{obs} & \cdots & \lambda_{foop}^m - \lambda_{obs} \end{bmatrix} \quad (10)$$

where $D_{rot}$ consists of the rotated positions of each FOP. $\varphi_{foop}^i$ and $\lambda_{foop}^i$ stand for the latitude and longitude locations of the $i$th point and $\varphi_{obs}$ and $\lambda_{obs}$ are the actual coordinates of the satellite observation. In equation 10, $\phi$ is the satellite instrument azimuth angle at observation location and $m$ is the number of FOPs. In order to get the final position of the IFOV, the rotated distances need to be added to the observation geolocation ($\varphi_{obs}$ and $\lambda_{obs}$) including the variation in distance between meridians

with latitude ($\cos(\varphi_{obs})$). Then IFOV ellipses of the footprint operator are aligned like the cross-track sensors measure them.

On Figure 3 the AMSU-A FOPs are visualised for a given part of a scanline inside the AA domain. The corresponding observations are shown for MHS FOPs for 5 neighbouring scanlines (Figure 4). Note that the pixels near the edges of the swath are also plotted on these figures, however, these pixels are generally blacklisted in the operational systems.

### 3.3 The emissivity retrieval in the footprint operator

Emissivity is the ratio of energy emitted from a natural material to that from an ideal blackbody at the same temperature. It varies with the frequency, with the surface type and also with the observation scan angle. The emissivity is an important component of the radiance data assimilation and difficult to properly determine especially over non-ocean surfaces. Over open ocean, the version 4 of the Fast Microwave Water Emissivity Model (FASTEM-4) is employed (Liu et al., 2011) where the final surface emissivity value of a given radiance data is internally computed in the RTTOV model. In the data screening, it

means that the open water emissivity is set by a two-step procedure and read/written into the observation database (ODB). Over land, the dynamic emissivity retrieval method (Karbou et al., 2006) is employed. An "effective" surface emissivity is retrieved from a window channel by reversing the radiative transfer equation. Short-range forecasts of atmospheric profiles and surface information are used as input to RTTOV (see Equation 11 below). The retrieved emissivity is then allocated to the adjacent sounding channels of the same instrument prior to the assimilation process as the following

$$\epsilon(\nu, \theta) = \frac{T_b(\nu, \theta) - T_a^u(\nu, \theta) - T_a^d(\nu, \theta)\tau}{(T_s - T_a^d(\nu, \theta))\tau} \quad (11)$$



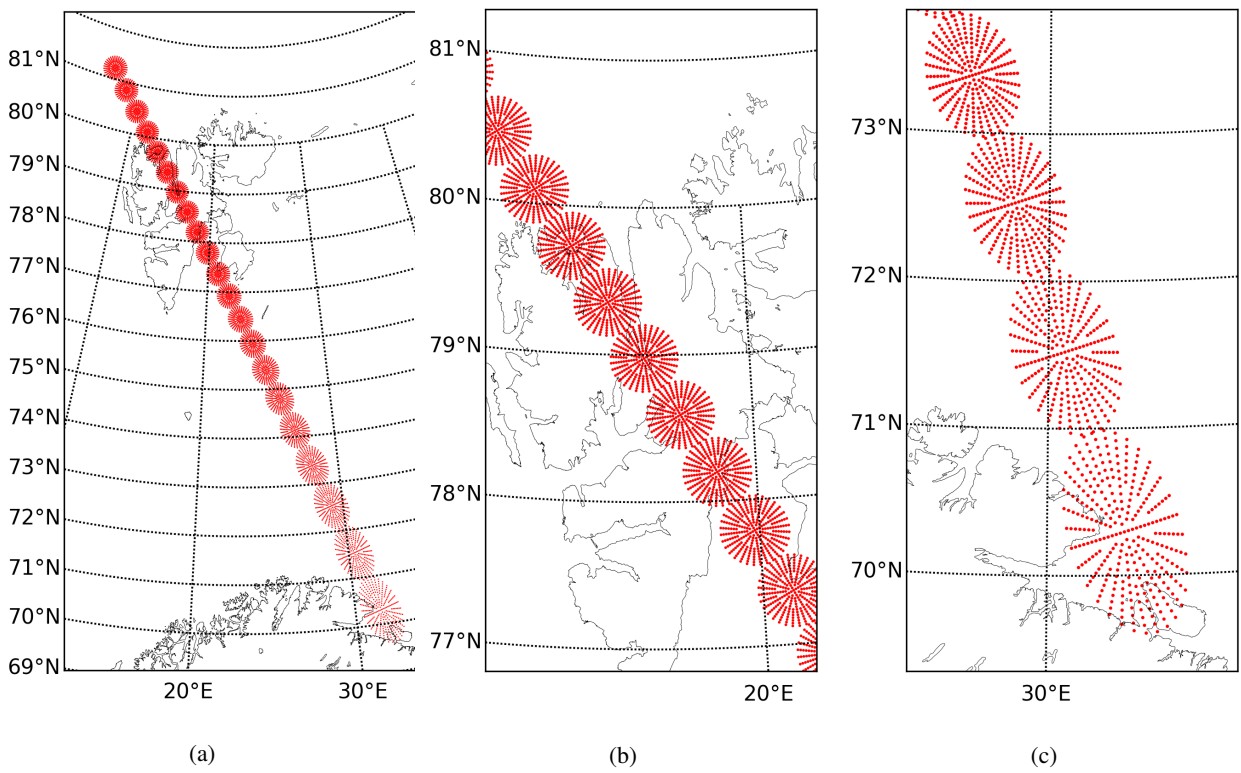

(a)                            (b)                            (c)

**Figure 3.** The FOPs for AMSU-A FOV ellipses on Metop-C satellite for one scanline (**a**), around the nadir (**b**), and near the edge of the swath (**c**). Sample taken from measurement on 20 March, 2020, 08:28 UTC.

where $\epsilon(\nu,\theta)$ represents the surface emissivity at frequency $\nu$ and at observation zenith angle $\theta$. $T_b(\nu,\theta)$ is the measured brightness temperature by the sensor, furthermore, $T_s$, $T_a^u(\nu,\theta)$, and $T_a^d(\nu,\theta)$ are the skin temperature and the atmospheric downwelling and upwelling temperatures respectively. In the AA operational configuration, channel 3 from AMSU-A and channel 1 from MHS are used to retrieve the land surface emissivity. Over sea ice, the surface emissivity is "reversed" at

channel 1 as it was retrieved at channel 2 to account for the frequency dependency (Karbou et al., 2014).

In the footprint operator, the final emissivity can be calculated in different ways depending on the choice of the averaging operation (see Section 3.1). For example, one option would be to retrieve one emissivity to be representative to the entire footprint or another - the chosen one of this study - is to derive emissivity values independently for each FOP (hereafter $\epsilon^i(\nu,\theta)$). On Figure 5a, the retrieved $\epsilon^i(\nu,\theta)$ over mixed surface conditions (i.e., sea ice, open ocean, and over land) can be

seen near the Svalbard archipelago for AMSU-A channel 6 data. Beside, on Figure 5b, the sea-ice chart created at MET Norway is presented over the same region for comparison. Near the coastal area of Svalbard, it can be seen that certain AMSU-A IFOVs can have two or even three different surface conditions within a single footprint (see the area of Wahlbergøya and Wilhelmøya, Svalbard). The larger retrieved emissivity values (red colour on Figure 5b) correspond to land and (thin) sea-ice surfaces, while low emissivity values (blue colour) correspond to open water.

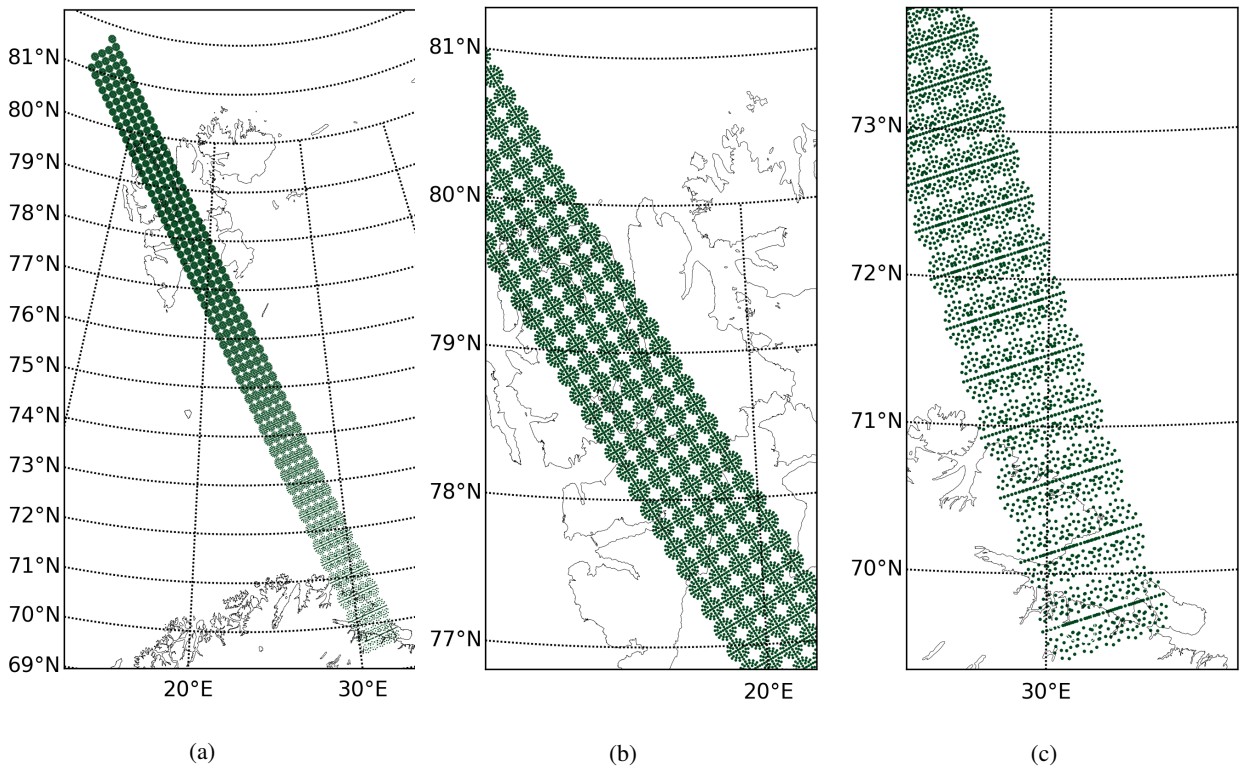

**Figure 4.** The FOPs for MHS FOV ellipses on Metop-C satellite for 5 scanlines (**a**), around the nadir (**b**), and near the edge of the swath (**c**). Sample taken from measurement on 20 March, 2020, 08:27 UTC.

One of the main technical challenges for this radiance observation operator is that the default system is "interactively" using the information stored in the ODB and not designed to handle for instance emissivity of the additional FOPs. In our implementation, the AMSU-A footprint operator uses roughly 300 FOPs for a single observation, increasing largely the computational cost of the radiance observation operator. The emissivity values of each FOP ($\epsilon^i(\nu,\theta)$) are stored in the extended ODB providing fast access to the subsequent assimilation tasks. It means that stored emissivity values can be invoked during the minimisation procedure ensuring the proper use of emissivity for the footprint operator. On Figure 6a, all $\epsilon^i(\nu,\theta)$ are shown for all available AMSU-A channel 6 observations with their corresponding FOPs for the same analysis as it was shown in Figure 5a. On Figure 6b, the $\epsilon^i(\nu,\theta)$ are also shown only for active data indicating the result after the data filtering procedure (basically blacklisting and thinning).

### 3.4 The footprint observation operator in the variational assimilation scheme

In the variational assimilation framework, the observation operator allows the computation of the model equivalent of the assimilated observation. In the footprint operator, the model equivalent is calculated for all operator points and the innovation (hereafter **d**) is obtained by the use of averaged model equivalents as follows

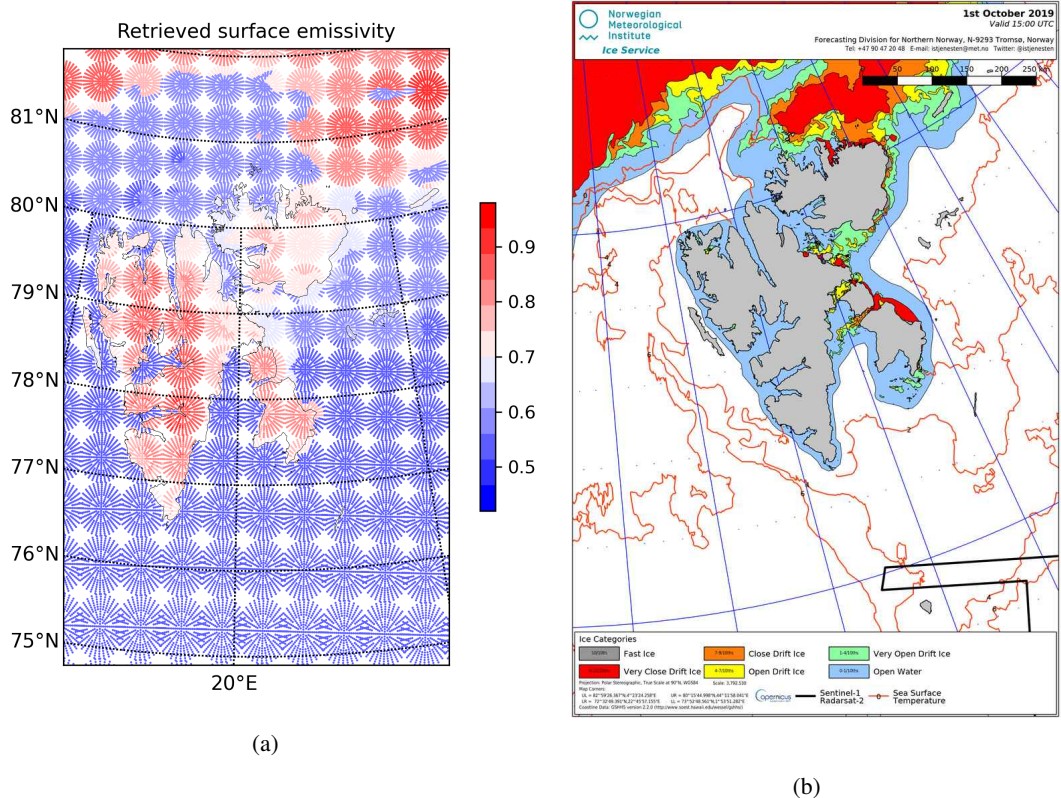

(a)

(b)

**Figure 5.** The retrieved surface emissivity of AMSU-A observations from NOAA-18 satellite by the radiance footprint operator with all FOPs (**a**). The sea-ice chart of MET Norway's sea-ice service showing various sea-ice categories (**b**) namely fast ice (grey), close drift ice (orange), very open drift ice (green), very close drift ice (red), open drift ice (yellow), and open water (blue). The sea surface temperature isolines are indicated by red lines. Data from 1 October, 2019 at 15 UTC are used.


$$\mathbf{d} = (\mathbf{y} - \overline{H[\mathbf{x}_b]}) \quad (12) \qquad \overline{H[\mathbf{x}_b]} = \frac{\sum_{i=1}^{m} H[\mathbf{x}_b^i]}{m} \quad (13)$$

where $x_b$ stands for the background state, $y$ is the observation, and $H$ is the observation operator. In the equation 13, $\overline{H[\mathbf{x}_b]}$ is the averaged model equivalent to be used in the assimilation and $m$ is the number of FOPs. It means that the $H[\mathbf{x}_b^i]$ model equivalents are averaged with equal weights in the current implementation of the radiance footprint operator. In the variational

assimilation scheme, a linearised observation operator and its conjugate transpose (so-called adjoint operator) are also needed. The footprint operator is built around the default radiance observation operator meaning that the adjoint footprint operator is basically the transpose of the linear averaging operator. In order to verify the correctness of the implemented adjoint operator,



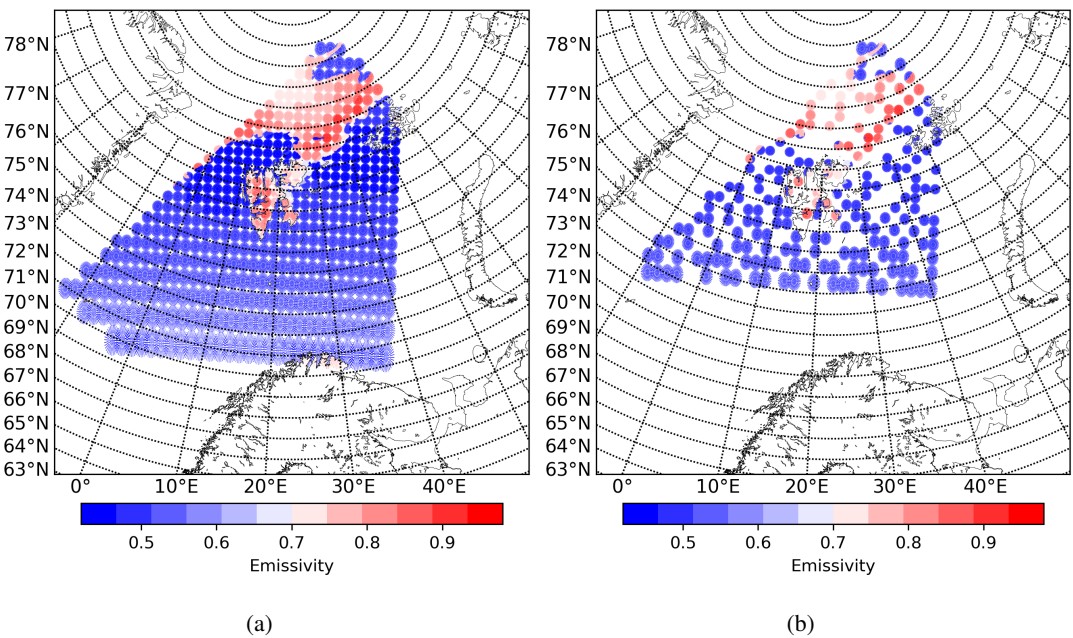

(a)                                              (b)

**Figure 6.** Maps of emissivity retrievals using all available AMSU-A channel 6 (**a**) and active (**b**) observations from NOAA-18 satellite, 1 October, 2019. at 15UTC using the radiance footprint operator.

the adjoint identity test was produced (see Table 1) and provided comparable results with the default radiance observarion operator (not shown).

| Adjoint identity | Footprint operator results |
|---|---|
| $< \mathbf{H}(\delta x), \delta y > =$ | -2.427335575862647232E+01 |
| $< \delta x, \mathbf{H}^T(\delta y) > =$ | -2.427335575862653627E+01 |
| Relative error = | 2.634528445688114808E-15 |
| The difference is | 11.9 times machine precision. |

**Table 1.** The results of the adjoint identity test for the footprint operator using NOAA-19 AMSU-A satellite radiances in the analysis of 9UTC, 20 March, 2020. $\mathbf{H}$ is the linear and $\mathbf{H}^T$ is the adjoint footprint operator, $<,>$ is the inner product, $x$ and $y$ are vectors of the space.

As it was already mentioned, the computational cost of the radiance assimilation is significantly increased when the footprint operator is switched on. Main reasons for this are that the RTTOV is called multiple times for a single observation and additional operations are needed through the ODB in order to make the proper emissivity retrieval for each FOP. In Table 2, a summary can be seen about the runtime performance of AA minimisation when the default and the footprint observation operator are applied for both AMSU-A and MHS data. The average speed of the minimisation is increased 22.5 and 75.7 % for MHS and





AMSU-A data respectively when the introduced footprint operator is employed. This result suggests that the optimisation is
essential before any real-time applications.

| Assimilation setup | Minim runtime performance | Relative increase |
|---|---|---|
| Default operator AMSU-A | 363 seconds | - |
| Footprint AMSU-A (301 FOPs) | 638 seconds | 75.7% |
| Default operator MHS | 342 seconds | - |
| Footprint MHS (45 FOPs) | 419 seconds | 22.5% |

**Table 2.** The mean runtime performance of AA minimisation in seconds using the default and the radiance footprint operator with AMSU-A and MHS data. Additionally, the relative increase of the runtime performance is indicated for the footprint operators. The average runtime performance was determined on the period between 1 and 31 January, 2021.

## 4   The simulated radiance data and its sub-footprint variability

We investigate the simulation of model-equivalent brightness temperatures generated by the radiance footprint operator independent of its use during the assimilation. The radiance footprint operator is efficient when the variability in model fields is
considerably large over the footprint area. In order to exhibit the variability in model fields, a case study with developing polar low (PL) inside the AA domain was investigated. In the morning hours (around 9 UTC) of 20 March, 2020, the developing PL was moving towards the Lofoten archipelago (see Figure 7a) causing severe weather events. At this time of the year, the sea-ice extent is substantial inside the AA domain (see Figure 7b showing the Fram strait and the Svalbard area) meaning that the chosen case study involved diverse meteorological and surface conditions for studying the radiance footprint operator and
the corresponding simulated data.

The variability in model fields is explored for the available AMSU-A and MHS data inside the AA domain. We compute the variability over the simulated brightness temperatures of the radiance footprint operator (using the FOPs) i.e., the standard deviation of $H[\mathbf{x}_b^i]$ for each satellite IFOV. On Figure 8a and 8b, the variability inside each AMSU-A footprint is shown for data from channel 1 and 7 respectively. The colour scales on Figure 8a and 8b are different and it allows to show only those IFOVs
where the standard deviation is the highest. It can be seen on Figure 8a that the sub-footprint variability likely corresponds to the variable surface conditions in case of radiance data from AMSU-A channel 1 (23.8 GHz window channel). On Figure 9a, the retrieved dynamical emissivity is shown (AMSU-A channel 1) indicating that the highest variability is over the sea-ice edge and the coastal area in northern parts of Scandinavia. The high variability in the observation equivalent is associated with the impact of mixed surfaces inside the IFOV (see also the area of Jan Mayen island and the corresponding two IFOVs).
High-peaking channels of AMSU-A are less or not sensitive to the surface, therefore, sub-footprint variability is not related to mixed surface conditions. On Figure 8b, the variability in model equivalents of AMSU-A radiances considering the 54.94 GHz channel (called channel 7) is plotted and the highest variability is obtained close to the edge of the satellite scan. It might be




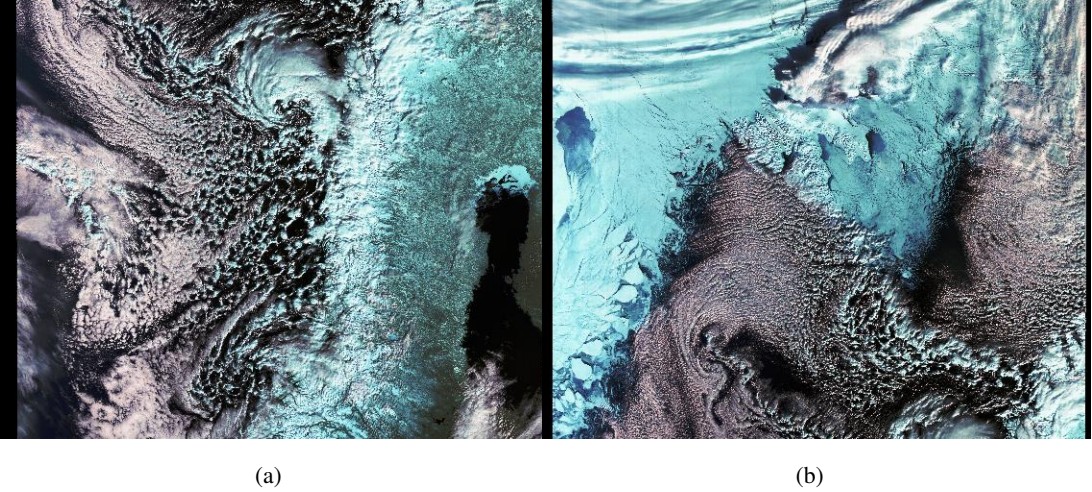

(a)                                                 (b)

**Figure 7.** Sentinel-3B satellite image captured on 20 March, 2020 at 10:16 UTC over the coast of Norway showing a PL (**a**) and Sentinel-3A satellite image taken on 20 March, 2020 at 10:52 UTC over Svalbard showing the sea-ice edge near the Fram strait and the south of Svalbard (**b**). Copernicus Sentinel data 2020, processed by European Space Agency.

explained by the fact that IFOV sizes are larger near the edges of the swath and the corresponding resolution gap is also larger between the model and the observation. On Figure 9b, the temperature field of the model background is shown suggesting that

there's small or no correspondence between the temperature gradient and the higher sub-footprint variability (Figure 8b).

On Figure 10a and 10b, the variability in model-equivalent brightness temperatures is shown for channel 1 and 3 of MHS radiances respectively. The colour scales are not the same on these figures indicating only the highest variability and the corresponding footprint. For window channel 1 of MHS (89.0 GHz frequency) similarly to the AMSU-A window channel 1, the standard deviation is the highest over mixed surface i.e., near the sea-ice edge and around coastal areas (Figure 10a). Mixed

surface scenes are also visible on Figure 11a showing all retrieved emissivity values ($\epsilon^i(\nu,\theta)$) of the FOPs for MHS channel 1. For the humidity sensitive channel (see Figure 10b), the highest variability is obtained over different locations and open ocean areas suggesting correspondence with atmospheric humidity conditions independently from the surface. On Figure 11b, the AA specific humidity field is shown around the same level where the MHS channel 3 is the most sensitive. Basically, the high variability over the MHS IFOVs on Figure 10b corresponds to the large variability and gradient in the humidity field (see

Figure 11b). It is worth highlighting that the PL area near the Lofoten archipelago has typically larger sub-footprint variability indicating small-scale variability near the centre of the PL (see a closer look of this area on Figure 12). In this situation, the use of the radiance footprint operator can be particularly important, although, this might require the all-sky framework for MHS radiance assimilation as well.

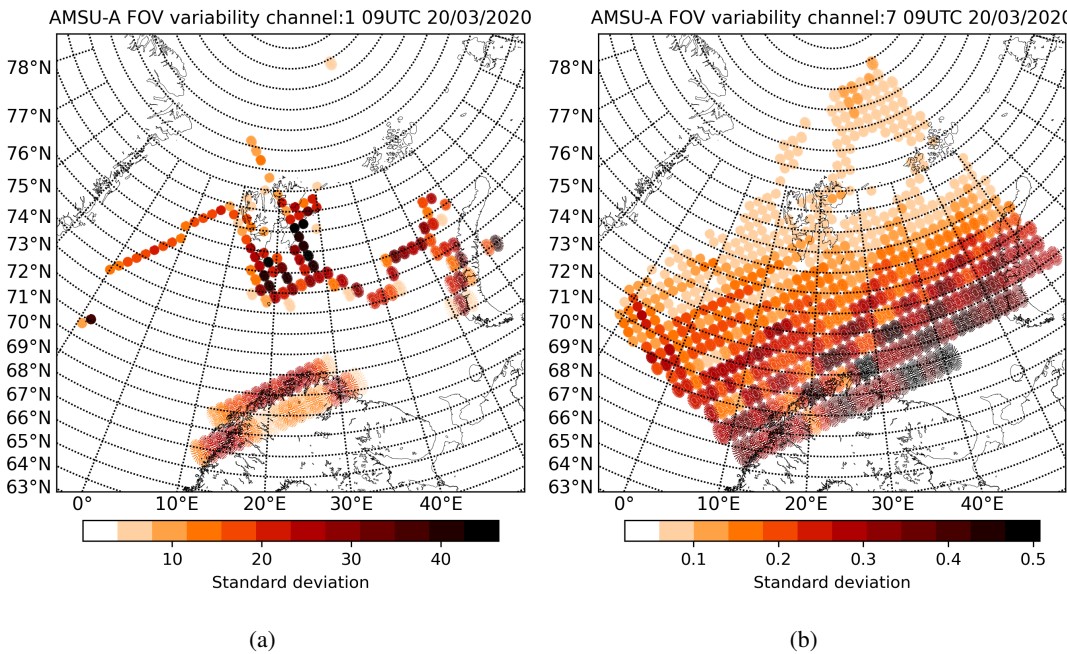

**Figure 8.** The variability in the simulated brightness temperatures inside the footprint of AMSU-A window channel 1 (**a**) and high-peaking channel 7 (**b**) from the NOAA-19 satellite on 20 March, 2020 at 9 UTC.

## 5 Departure-based diagnostics

In this section, OmB departures are examined comparing the statistics of the default and the footprint observation operators for AMSU-A and MHS radiances. The diagnostics are assessed on a one-month and a two-month period for AMSU-A and MHS respectively excluding short spin-up intervals. For MHS radiances, a longer diagnostic period was necessary to collect enough data inside AA domain (i.e., amount of data per the scanning position between 1000 and 4000). The radiance observations from Metop-B, Metop-C, NOAA-18, and NOAA-19 satellites are used whitelisting pixels near the edges of the swath in the AA assimilation runs. Apart from the applied observation operators, the default assimilation settings, namely, the predefined observation and background errors, thinning distances, and the quality control procedure were the same in both assimilation experiments. On Figure 13a, the standard deviation of OmB departures is plotted using all operationally utilised clear-sky AMSU-A data during the studied period. It can be seen that the use of the radiance footprint operator provides small but consistent reduction in the standard deviation of OmB departures. The same can be concluded from the standard deviation of MHS departures which is plotted on Figure 13b.

On Figure 14a and 14b, the standard deviation of OmB departures for AMSU-A channel 5 and 9 are shown as a function of satellite scanning positions. By the use of the radiance footprint operator, the reduction in OmB statistics for channel 5 and 9 is apparent near the edges of the swath and also for smaller zenith angles. It means that taking into account the satellite footprint might help to reduce the difference between observations and model equivalents in the full swath. The reduction is slightly

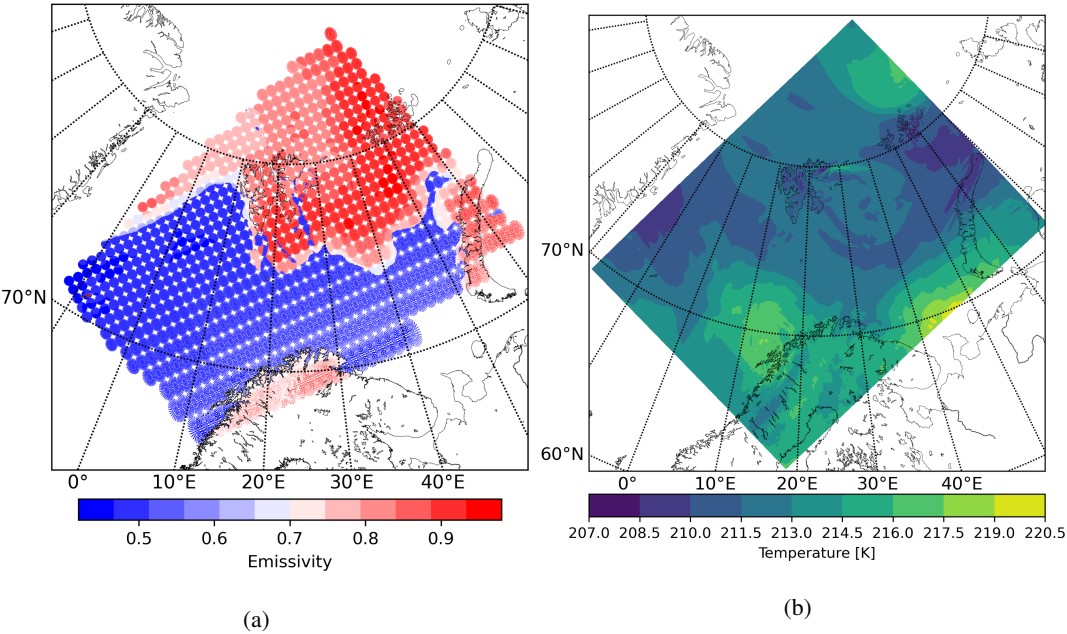

(a)        (b)

**Figure 9.** The retrieved dynamic emissivity of AMSU-A channel 1 (**a**) from the NOAA-19 satellite using the radiance footprint operator and showing all emissivity values $\epsilon^i(\nu, \theta)$ of the FOPs. The AA temperature field (**b**) at level 15 (around 9 km altitude) on 20 March, 2020 at 9 UTC.

larger near the edges for high-peaking channels which might be explained by the larger representation error for the outermost scan-positions. On Figure 15a and 15b, the normalised standard deviation of OmB for AMSU-A channel 5 and 9 are shown. For low-peaking channel, it can be seen that the reduction is considerable for pixels close to the center of the swath. Additionally, the number of OmB departures are plotted on these figures showing that thinning procedure filters more observations at nadir (higher data density) compared to observations near the edges of the swath.

Concerning MHS radiances, the standard deviation of OmB departures for MHS channel 4 and 5 are shown as a function of 90 scanning positions (see Figure 16a and 16b). When the MHS footprint operator is applied, the reduction in OmB statistics is noticealbe for most of the pixels in the full satellite swath. For this sensor, a longer period was necessary to run in order to gain larger statistical sample. On Figure 17a and 17b, the normalised standard deviations of OmB for MHS channel 4 and 5 are also shown and can be seen that the reduction is perceptible for most of the MHS scan positions. Similarly to AMSU-A

Figure 15a and 15b, the number of OmB departures are also plotted on these figures.

## 6   Conclusions and future outlook

The footprint representation of microwave radiance observations was explored in the mesoscale AA data assimilation system. The resolution gap between the data and the NWP model is apparent, especially for AMSU-A radiances near the edges of

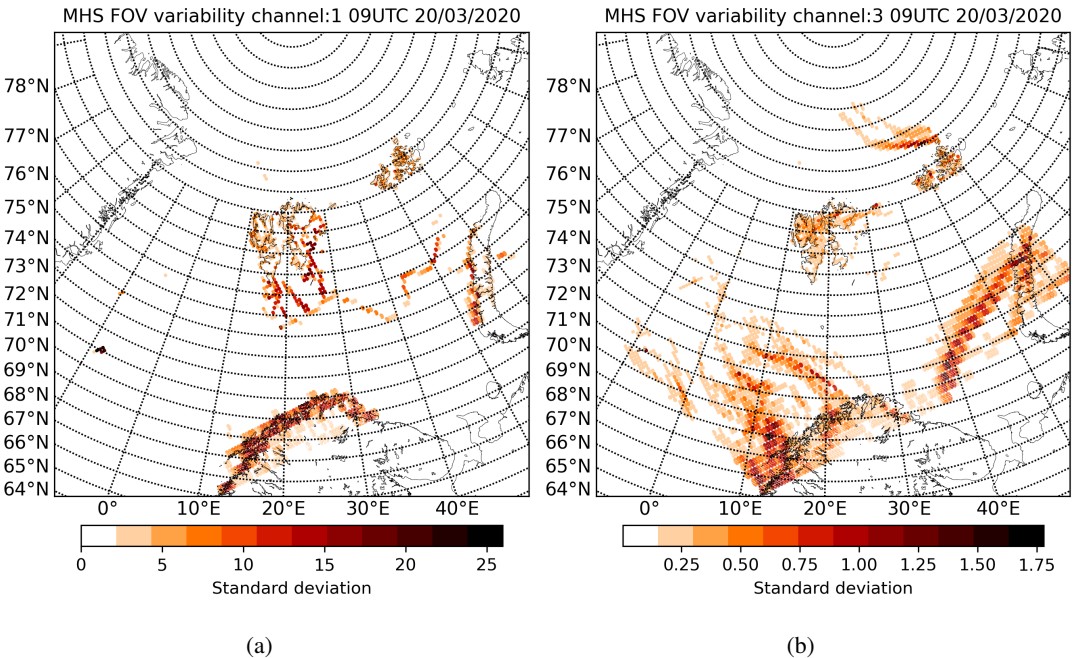

(a)                                          (b)

**Figure 10.** The variability in the simulated brightness temperatures inside the footprint of MHS window channel 1 (**a**) and channel 3 (**b**) from the NOAA-19 satellite on 20 March, 2020 at 9 UTC.

the satellite scan. The use of the footprint operator helps to make the data assimilation system more optimal and to avoid

constraining unobserved scales and processes in the mesoscale analysis. The implementation of the footprint operator can follow different strategies. In this paper, the averaging operation after the RTTOV simulations was explored making the implementation computationally more expensive and more appropriate in the non-linear radiance observation operator. Such an implementation in the AROME variational assimilation system involves technical challenges and increased computational costs. For instance, the storage and the proper utilisation of retrieved dynamic emissivity values are challenging tasks in the

minimisation procedure for all footprint operator points independently. Beyond the technical implementation, the potential benefit of the radiance footprint operator was investigated in a particular case study exploring the sub-footprint variability of the simulated data. The added value of the radiance footprint operator is expected to be negligible over areas of homogeneous meteorological conditions and its use is more important where the variability in model fields is considerably larger. In this case study, we demonstrated that the variability is higher near the edges of the swath (large footprint size) for a high-peaking

AMSU-A channel and the high variability also corresponds to particular weather phenomena like a PL in the case of simulated MHS brightness temperatures. Additionally, departure-based statistics were investigated for AMSU-A and MHS radiances showing small, but consistent reduction in the standard deviation of OmB.

     Regarding future outlook, the implemented radiance footprint operator can be further developed. For example, the footprint representation helps to take into account heterogeneous conditions inside the IFOV area and to make a better quality control



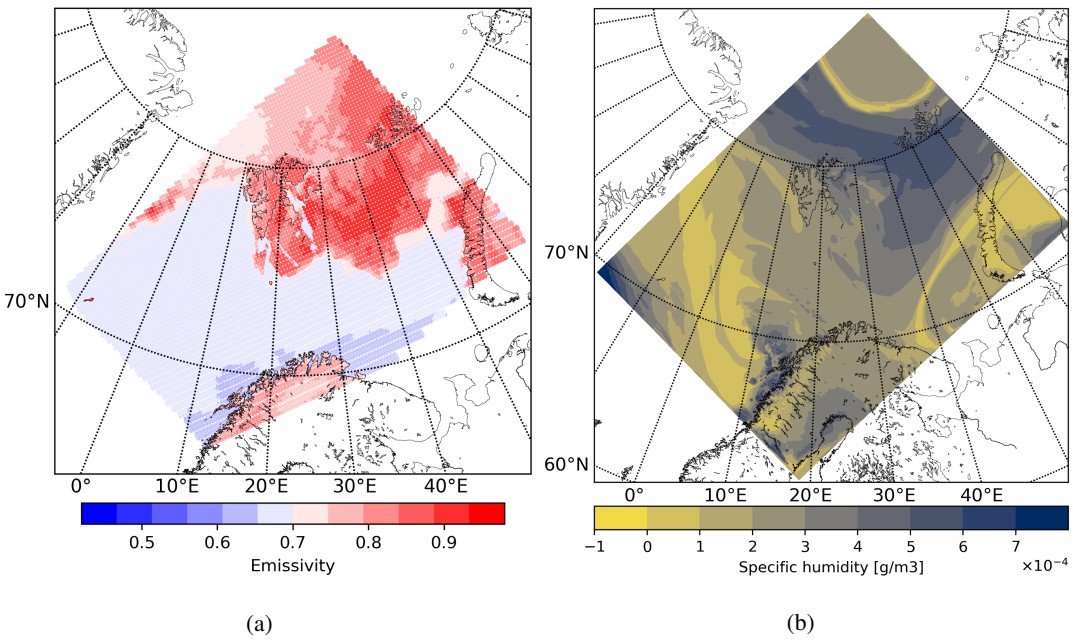

**Figure 11.** The retrieved dynamic emissivity of MHS channel 1 (**a**) from the NOAA-19 satellite using the radiance footprint operator and AA specific humidity field (**b**) at level 30 (around 3.6 km altitude) on 20 March, 2020 at 9 UTC.

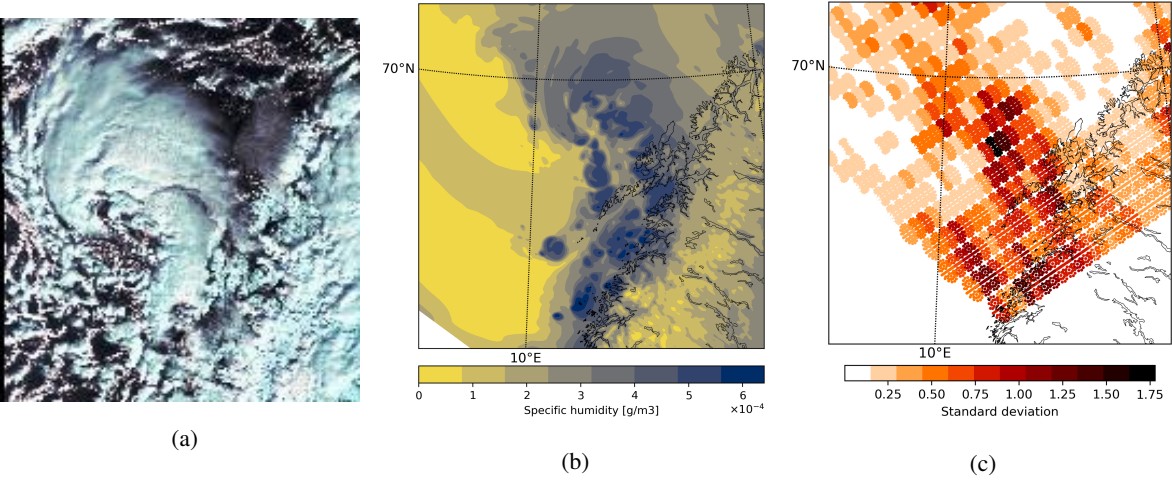

**Figure 12.** Sentinel-3A satellite image captured on 20 March, 2020 at 09:14 UTC showing the PL near the Lofoten archipelago (**a**). Copernicus Sentinel data 2020, processed by European Space Agency. AA specific humidity field at level 30 (around 3.6 km altitude) over the Lofoten archipelago (**b**) on 20 March, 2020 at 9 UTC. The variability in the simulated brightness temperatures inside the footprint of MHS channel 3 from the NOAA-19 satellite on 20 March, 2020 at 9 UTC (area over the Lofoten archipelago) (**c**).



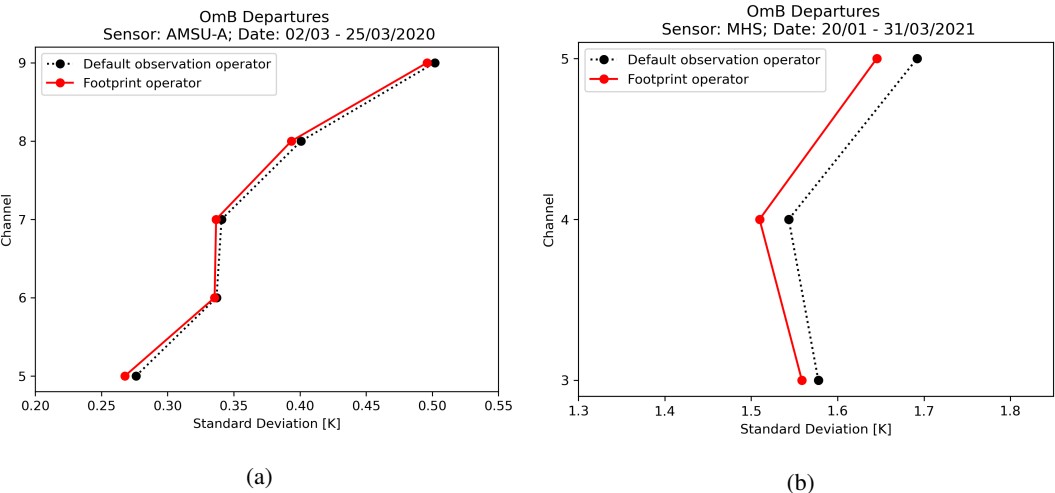

(a)                                                      (b)

**Figure 13.** The standard deviation of OmB for AMSU-A channels (**a**) and the standard deviation of OmB for MHS channels (**b**). The black dashed lines show the statistics of the default radiance observation operator, and the red solid lines are the results of the radiance footprint operator. The statistics were accumulated during the period between 2 and 25 March, 2020 for AMSU-A data and the period between 20 January and 31 March, 2021 for MHS departures.

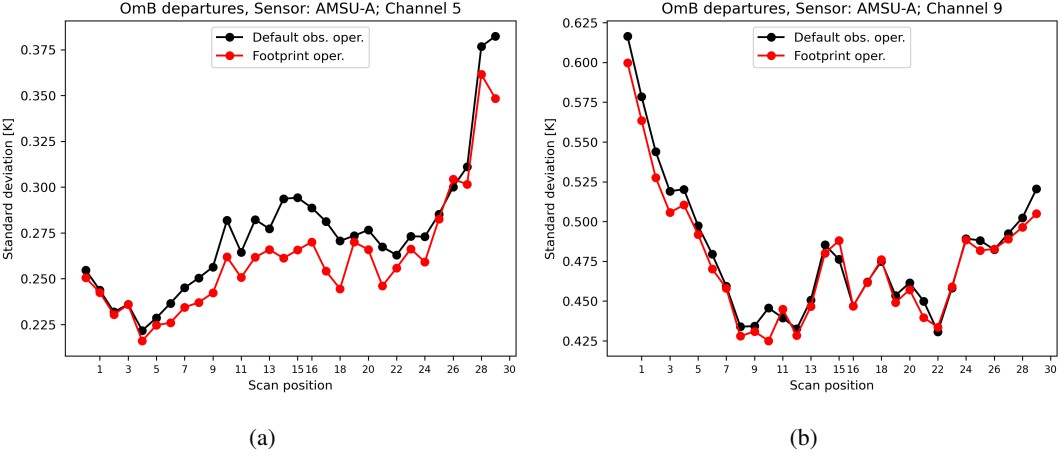

(a)                                                      (b)

**Figure 14.** The standard deviation of OmB using data of AMSU-A channel 5 (**a**) and channel 9 (**b**) as a function of satellite scanning position. The black line stands for the default and the red line is for the footprint operator respectively. The statistics were computed on a period between 2 and 25 March, 2020.

for the radiance observations. Additionally, it might help to better assimilate radiances in the all-sky framework e.g., to better describe cloud and rain inside the footprint area. The radiance footprint operator might also help to start using satellite observations that are sensitive to the surface i.e., assimilating radiance over all-surface conditions. The implemented radiance footprint operator uses interpolated FOPs and corresponding vertical profiles neglecting that the instrument has a slanted path

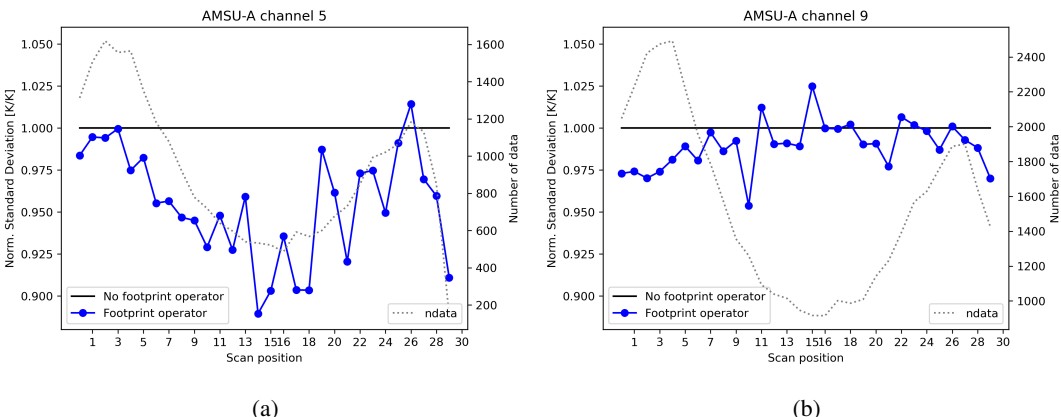

(a)                                                    (b)

**Figure 15.** The standard deviation of OmB departures from the footprint operator normalised by the statistics from the default operator (blue line) for AMSU-A channel 5 (**a**) and channel 9 (**b**) as a function of satellite scanning position. The number of observation minus background differences are also shown with grey dashed line. The statistics were computed on a period between 2 and 25 March, 2020.

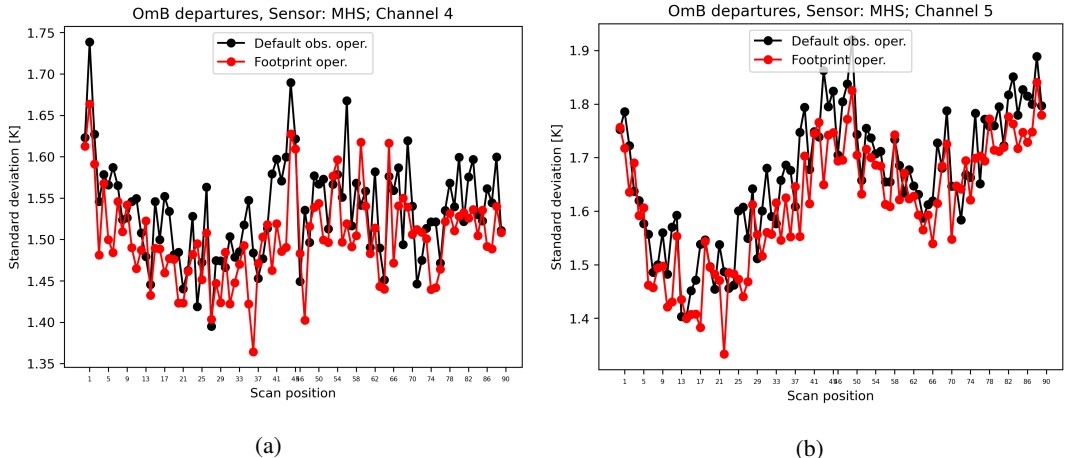

(a)                                                    (b)

**Figure 16.** The standard deviation of observation minus AA forecasts of MHS channel 4 (**a**) and channel 5 (**b**) as a function of satellite scanning position. The black line stands for the default and the red line is for the footprint operator respectively. The statistics were computed on a period between 20 January and 31 March, 2021.

through the atmosphere. Therefore, the combination of the slant-path and the footprint operator can be considered as a future
development improving even further the use of radiance data near the edges of the swath. Furthermore, the introduced footprint operator assumes equal contributions from each FOP and therefore, applies a boxcar averaging function. However, this might be oversimplified for radiance measurements where the radiance signal strength does not show a uniform distribution and the satellite footprint is basically defined by the half power beam width. It means half the energy that part of the main lobe where the power is below 3 dB plus side lobes comes from area (main lobe and side lobes) outside the footprint. In the future
development of the radiance footprint operator, the signal stength, the effective field-of-view, and the antenna pattern need to



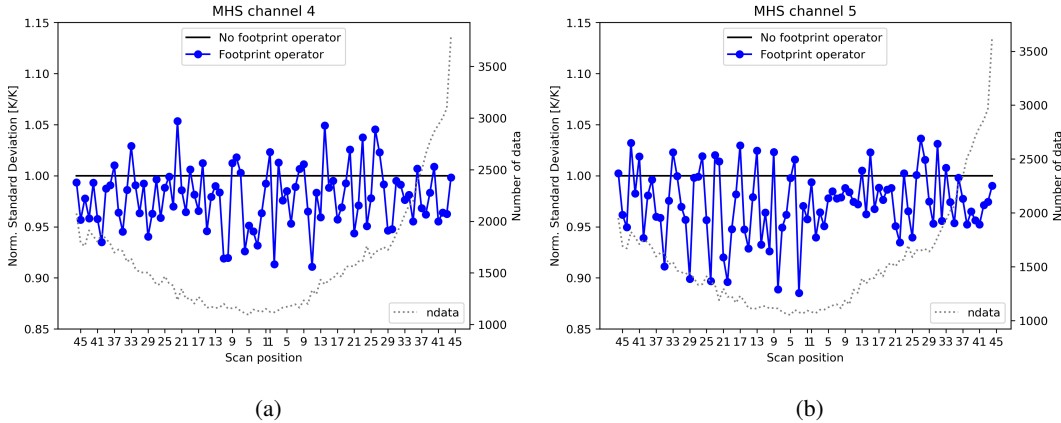

(a)                                             (b)

**Figure 17.** The standard deviation of OmB departures from the footprint operator normalised by the statistics from the default operator (blue line) for MHS channel 4 (**a**) and channel 5 (**b**) as a function of satellite scanning position. The number of observation minus background differences are also shown with grey dashed line. The statistics were computed on a period between 20 January and 31 March, 2021.

be considered as well. Beyond the improved representation of the radiance footprint, the optimisation is also important for the cost-efficient application. The trade-off between accuracy and cost-efficiency is particularly important for future operational use.

*Code availability.* The AROME-Arctic assimilation scheme, the implemented footprint operator, and HARMONIE-AROME codes, along
with all their related intellectual property rights, are owned by the members of the ACCORD (A Consortium for COnvective-scale modelling Research and Development) consortium. This agreement allows each member of the consortium to license the shared ACCORD codes to academic institutions of their home country for noncommercial research. Access to the codes of the ACCORD System can be obtained by contacting one of the member institutes by submitting a request in the Contact link below the page of the ACCORD website (http://www.umr-cnrm.fr/accord/) and the access will be subject to signing a standardized ACCORD license agreement. The observation dataset (including the
departure-based statistics) has been packed and available for download at Zenodo (https://doi.org/10.5281/zenodo.10851015, Mile (2024)).

*Author contributions.* MM and RR designed the footprint observation operator and MM developed the model code. SG instructed the implementation of dynamical emissivity retrieval in the footprint operator. MM designed the experiments and MM carried them out. MM prepared the manuscript with contributions from all co-authors

*Competing interests.* The contact author has declared that none of the authors has any competing interests.






*Acknowledgements.* First of all, the authors are grateful to Pascal Brunel for the guidance of the radiance footprint calculations. Additionally, we are grateful for the ECMWF and Météo-France colleagues making the basis of 2D-GOM-arrays. The authors kindly acknowledge the support of Norwegian Research Councils through the Arctic weather prediction project called ALERTNESS (project no. 280573) and the support of High Performance Computing Facility at ECMWF which resources were provided via a special project called spnomile.



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
