# Peer review of "Exploring the footprint representation of microwave radiance observations in an Arctic limited-area data assimilation system"

_Geoscientific Model Development, 2023_

## Author Comment (AC1)

Author's response for the manuscript titled
„Exploring the footprint representation of microwave radiance observations in an Arctic limited-area data assimilation system"

First of all, the authors would like to thank the comment of the Anonymous Referee #1! We considered the comments and revised the manuscript accordingly. All the changes have been highlighted in the manuscript to ease the evaluation of the changes. In this letter, the answers are written after each reviewer's comments (with blue colour).

Sincerely,

Máté Mile

On behalf of all authors

*Radiance assimilation usually assumes point observations, which is fine for global NWP models, while for high resolution regional NWP models (and future high resolution global NWP models), the footprint size is too large and the footprint operator needs to be considered together with observation operator (RTM). The manuscript provides a comprehensive analysis and discussion on how such footprint operator can be developed and used. I have just one question and one suggestion.*

*Question: why the grided brightness temperature averaging is a straightforward one? is it possible to apply spatial response functions of AMSU/HMS channels? The standard deviation of OmB seems overestimated if the spatial response functions are not considered.*

Indeed, this aspect is partly discussed in the conclusion section. In the current implementation, the footprint operator aims a near equidistant sampling inside the FOV in order to match (or come close to) the model horizontal resolution. However, as the reviewer suggests, such sampling approach with equal-weight averaging does not take into account the antenna pattern of the employed microwave instruments.

In order to consider the antenna pattern in the footprint operator, we have started to test two approaches. One idea is to apply weighted averaging keeping the implemented sampling and ensuring larger contributions for the operator points near the bore-sight and less weights towards the edge of the IFOV. See an example figure below showing the AMSU-A footprint operator points with the assigned weights of the averaging.

[Figure]

Figure 1. Taking into account radiance antenna pattern by weighted footprint operator points inside the AMSU-A IFOVs.

The statistics of OmB (observation minus background) departures were collected comparing the equal-weight and the weighted averaging footprint operator performance using MHS radiance observations. On the figure (figure 2.) below, this comparison can be seen for a statistics of a short period suggesting that the representation of the antenna pattern (i.e., the weighted averaging approach) is less meaningful and provides very similar reduction in OmB standard deviation than the equal-weighting approach. This approach is still under evaluation and the assigned weights might not be fully appropriate for the antenna response.

[Figure]

Figure 2. The standard deviation of OmB departures of MHS footprint operator normalized by the default radiance observation operator using radiance data from MHS channel 4. Here, we compare the footprint operator with (red line) and without (green) the antenna pattern representation with the weighted average approach.

Additionally, it's worth mentioning that the MHS footprint representation (in the figure above for both runs) covers the effective FOV area and therefore not comparable with the results of Figure 17. in the manuscript.

Another idea is to keep every footprint operator point equally important (i.e., equally-weighted), but to apply oversampling near the bore-sight and to have fewer and fewer points towards the edge of the FOV. It would result in a certain representation of the antenna pattern by taking into account model information with larger relative weight or importance near the bore-sight. On the figures below, the different sampling strategy can be compared for AMSU-A and MHS pixels.

[Figure]

a)   b)

Figure 3. Different sampling in the microwave footprint operator. Figure a) shows AMSU-A pixels with the sampling used in the manuscript (left pixel) and a different more dense sampling scheme around the bore-sight (right pixel). Figure b) shows the same, however, new sampling is on the left and original sampling from the manuscript is on the right for MHS pixels.

The impact of this new and different sampling shows no further reduction in the standard deviation of OmB departures (not shown). However, this scheme needs further evaluation since the number of operator points are also changed.

These above-mentioned findings are preliminary and experimental ones, therefore, we have not incorporated them into the revised version of the manuscript. There is an ongoing work that will study an optimized version of the microwave radiance footprint operator with improved runtime performance and accuracy in an observing system experiment. In the current revised manuscript we have extend the conclusion with one sentence to explain this a bit further.

*One suggestion: it worth noting a related research using high spatial resolution AHI observation for studying the IR sounder sub-footprint moisture variation, Di et al. (2021) found that the current IR sounders (such as CrIS, IASI, GIIRS etc.) with spatial resolutions between 12 and 16 km have typical average sub-footprint brightness temperature variations (BTVs) between 0.8 and 1.5 K over land, a 1 K variation in 6.25 μm water vapor absorption band corresponds to a 10% – 20% upper tropospheric moisture variation depending on the atmospheric humidity. Such sub-footprint BTVs, without being accounted for, may introduce additional uncertainties in quantitative applications*

*such as radiance assimilation. Their study provides another evidence on the needs of footprint operator in data assimilation for high resolution NWP models.*

*Di, Di, Jun Li, Zhenglong Li, Jinlong Li, Timothy J. Schmit, and W. Paul Menzel. "Can current hyperspectral infrared sounders capture the small scale atmospheric water vapor spatial variations?." Geophysical Research Letters 48, no. 21 (2021): e2021GL095825.*

Thank you for this reference that is certainly relevant for the subject. It is now mentioned in the Introduction section of the revised manuscript.

---

## Author Comment (AC2)

Author's response for the manuscript titled
„Exploring the footprint representation of microwave radiance observations in an Arctic limited-area data assimilation system"

First of all, the authors would like to thank the comment for Dave Duncan! We considered the comments and revised the manuscript accordingly. All the changes have been highlighted in the manuscript to ease the evaluation of the changes. In this letter, the answers are written after each reviewer's comments (with blue colour).

Sincerely,

Máté Mile

On behalf of all authors

*The authors present a method to account for sub-FOV features in the forward model of a regional data assimilation system, applied to ATOVS microwave radiances in the Arctic. This paper is very well-written and relevant for publication in this journal. It is clearly laid out and was a pleasure to read through.*

*Minor revision is recommended to address some small clarifications. Two larger comments are offered for discussion and are not to be viewed as requirements of the revision, but could be addressed by the authors if they choose. These could also be viewed as potential avenues for future work, alongside the already-identified areas such as slant-path RT and accounting for the antenna response. The more general comments are given first, followed by the more detailed points by line number:*

- *The analysis focuses more on the surface emissivity inhomogeneities that can exist within a microwave FOV, but it is interesting to see that even higher-peaking channels like MHS-4 and non-surface-sensitive AMSU-A channels also exhibit reductions in OmB. This would suggest that sub-FOV inhomogeneities in the water vapour field (and possibly temperature) are significant, as has been suggested by some previous studies (e.g. https://doi.org/10.5194/amt-11-6409-2018) though of course a larger effect for all-sky simulations (https://www.mdpi.com/2072-4292/12/3/531). It would be interesting to quantify this, i.e. what is the contribution of reduced OmB from better surface representation vs. the atmospheric inhomogeneity. Whether or not the authors choose to elaborate on this aspect, it might be worth a little more discussion of the atmospheric element of sub-FOV inhomogeneity, particularly for Fig. 12 as it is currently not discussed much. It's also maybe worth mentioning that sub-FOV inhomogeneity can be quite important for lowest frequencies especially such as for SST (e.g. Fig 2 here: https://amt.copernicus.org/articles/12/6341/2019/), and will be a key concern for using CIMR in LAMs going forward.*

  First of all, the authors would like to thank Dave Duncan for the interesting comment. We were uncertain how to investigate it and how to determine practically the various contributions in the reduced OmB departures. We are open to follow the discussion on this topic and please find below a few tests that might be related to the reviewer's question without proving the actual answer unfortunately.

On a different domain (MetCoOp – covering Scandinavia), we checked the raw model temperature fields at higher model levels in order to assess the variability in temperature fields in 50km and 150km segments. See figure 1 showing the model temperature fields in the top row and the corresponding standard deviations below. The model levels 5, 10, and 20 corresponds to 16.9, 12, and 6.9 km standard atmosphere heights (taken from the HARMONIE-AROME forecast setup).

[Figure]

Figure 1. The HARMONIE-AROME (MetCoOp domain) temperature model fields at level 5, 10, and 20 on subplot a), b), and c). The standard deviation of model temperature fields in 50 km segments (grid-boxes) at level 5, 10, and 20 on subplot e), f), and g) respectively. The variability in 150km segments (grid-boxes) at level 5, 10, and 20 on subplot h), i), and j) respectively.

It doesn't bring us much closer to the answer, but it demonstrates the variability in model fields (without RT) and we can see that certain atmospheric variability exist on high model levels as well when we consider AMSU-A IFOV sizes.

Additionally, we tried to select short periods with different weather regimes (high-pressure and low-pressure systems inside the AROME-Arctic domain). Atmospheric inhomogeneities might be larger in dynamically active weather situations, therefore, we expected to see

greater reduction in OmB standard deviation by the use of the footprint operator for high-peaking channels. On the figure below, two periods were compared for OmB standard deviations with (Fop) and without (Def) the AMSU-A footprint operator. The first period (call HIGHPRES) is between 19 and 23 February and the second (LOWPRES) is between 11 and 14 February, 2021. It's worth mentioning that all observations inside the AROME-Arctic domain were considered in these statistics and dominant weather patterns (Low-pressure, high-pressure) were not always covering the entire domain. This might impact the statistics below.

[Figure]

Figure 2. OmB standard deviations as a function of AMSU-A channels. Figure a) shows the different periods (LOWPRES – dashed; HIGHPRES – solid lines) and different runs (No footprint operator – black; Footprint operator – red lines). On Figure b), the difference between the default and the footprint observation operator can be seen.

One can see that the reduction in OmB is somewhat higher in the LOWPRES period, but it is not consistent and not restricted to high-peaking channels, therefore, it is difficult to conclude anything exact from these results. Any comments or further suggestions are welcome in this subject.

Regarding the discussion of sub-footprint inhomogeneity, the conclusion section in the manuscript has been extended shortly to mention this.

- *The FOP spacing is justified in section 3.2, implemented at roughly the model resolution of about 2.5km, and of course there is computational cost to increasing the RT complexity. Did you test finer or coarser spacing of FOPs? The 'trade-off' mentioned in the last sentence of the conclusions is an important one for operational DA, and it might be worth quantifying whether 2.5km spacing is sufficient, or for example if 5km spacing still reduces OmB significantly but is much cheaper. Along the same lines, while Figs. 13-17 contain a lot of useful information, a simple statement of total OmB reduction for a characteristic channel (e.g. 4% for AMSU-A-5 or whatever) would be a good headline value to include either in the main text or even the abstract as a way to demonstrate the significance of your results.*

Different sampling resolutions were tested on a short period with AMSU-A data only. The introduced sampling consists of 301 FOPs in the manuscript and we have calculated OmB

statistics with 77, 205, and 621 FOPs (so both under- and oversampling of the model resolution). On the figure below, it can be seen that with 77 FOPs (which corresponds to around 5 km sampling resolution in the footprint operator) provide similar amount of reduction in the standard deviation of OmB. It suggests that optimization with less number of FOPs can provide actual improvement in the runtime performance achieving similar reduction of the representation error. We plan to make an observing system experiment in the near future where the footprint operator is activated for multiple microwave sensors and optimized for possible operational implementation. However, it is not planned to include these optimization results in the manuscript.

Indeed, it is a good suggestion to mention a total OmB reduction in the text, so it is now mentioned both in the abstract and in the conclusion section as well.

[Figure]

Figure 3. Different sampling in the microwave AMSU-A footprint operator showing default operator (black), footprint operator with 301 (blue line), with 77 (green), with 205 (purple), and with 621 FOPS (yellow). Figure a) shows OmB standard deviations as a function of scan positions for AMSU-A channel 5. Figure b) shows the same for AMSU-A channel 9 data. Figure c) and d) show the normalized standard deviations for the same channels and with the different sampling.

*L13 Does AROME not use channels 10-14 on AMSU-A? Or maybe these were also improved with the footprint representation? If not, consider changing to 'tropospheric channels' to clarify.*

AMSUA-A channels 10-14 are not used in AROME-Arctic, so it has been clarified to the abstract.

*L21 'can be mentioned'? Not sure if this is a typo or it should be reworded, but it was unclear what this meant.*

This sentence has been rephrased in the manuscript.

*L30 'the sensitivity to the surface is relatively small.' This is an odd wording. I'd suggest removing 'due...small' because plenty of surface-sensitive channels are also assimilated at many centres using FASTEM, such as the MW imagers.*

Indeed, this sentence has been modified and the second part of it has been removed.

*L43 Probably okay to just cite one of the Bormann papers on slant-path RT*

The Tech. Memo. reference has been removed.

*L49 Maybe worth mentioning that radiance assimilation is much more effective in the Arctic during summer (Lawrence et al. 2019), presumably because complexities such as sea-ice edge and snow-cover are detrimental to optimal assimilation of MW radiance in the Arctic wintertime, which is your study period.*

First of all, we would like to thank for the interesting suggestion. Indeed it is worth mentioning and this reference has been added. However, we think a limited-area study is also important to mention here. Randriamampianina et al. (2021) showed that the impact obtained by the microwave radiance is actually larger in the winter and it might be explained that the global model (via lateral boundary conditions) allows the LAM system to gain larger impact during winter by microwave radiance data assimilation.

*L50 Suggest replacing 'inadequately' with the simpler 'not'*

It has been replaced in the manuscript.

*L76-83 It would be good to cite where these values came from. Perhaps official NOAA documentation?*

References (NOAA and EUMETSAT) have been added to the manuscript.

*L104 Not sure what 'increased assimilation cycle frequency' means here? Please reword.*

This sentence has been rephrased in the manuscript.

*L121 Please reword 'the neglected effects of the incorporated RTTOV in the observation operator' – seems like it might be missing a word.*

The averaging in model space would assume that RT is linear in the footprint operator, however, the formulation of this sentence was not appropriate. Therefore, it has been removed from the manuscript.

*L160 What is meant here by 'Earth's frame'? Is that the surface or something else?*

Yes, it's Earth's surface that has been corrected in the manuscript.

*L190 Is the subscript 'foop' a typo here for 'fop'?*

Indeed, it is an index for footprint operator points, so 'FOP' (capital letters) is the correct one and has been modified in the manuscript.

*L220 It's a little confusing to say that it's the retrieved emissivity for channel 6 in Figure 5, as presumably it is the surface-sensitive channel 3 that's used for the emissivity retrieval of 50+ GHz channels; the same comment applies for L272, where I would've expected ch3 to be used rather than ch1 for the emissivity retrieval.*

These sentences are reformulated in the manuscript in order to make them clear.

*L222 Unfortunately most readers won't know where Wahlbergøya and Wilhelmøya are within Svalbard, so if you refer to them in the text it would be helpful to label them on the map.*

An index figure has been added to Fig4a in order to highlight this area.

*Fig 5 If it's not too hard to do, rotation of panel (a) so that it matches the projection of panel (b) would facilitate more direct comparison of the two panels for readers.*

Fig 5 has been updated in the manuscript.

*Table 2 Are these performance values for the whole system minimisation (i.e. including full observing system) or only considering these instruments? Good to clarify this as it certainly will impact readers' interpretation of the computational cost increase.*

It has been corrected and stated now clearly in the manuscript.

*Fig 7 It would be helpful to add lat/lon values on these panels if possible.*

Unfortunately, adding lat/lon information is not possible, since the figures were produced by ESA. So the figures have not been updated in the manuscript.

*Fig 8 Just a suggestion, but it might be nice to see a third panel in the middle with channel 5, as this might show a combination of the two sensitivities.*

Thanks for the good suggestion, a third figure with channel 5 variability has been added.

*Fig 9 Would it be possible to combine Figs 8 & 9 into a four-panel plot? I found myself flipping back and forth to compare the two. The same applies to Figs. 10 & 11.*

Yes, good idea, these figures are now combined in the new revised manuscript.

*L295 Is this a cycling DA experiment, with bias correction etc. evolving in time? I would presume so as spin-up is mentioned.*

Yes, this is the case and both experiments have been initialized by the same VARBC coefficients that are more appropriate for the default observation operator runs. So a spin-up period is applied enabling adaptation of bias coefficients for the footprint operator runs.

*Fig 10 The same comment as for Fig 8 – adding a panel with MHS-5 would be interesting to show the mix of surface and atmospheric sensitivity.*

A figure with MHS channel 5 has been added into the manuscript.